# Assessing Pragmatic Skills in People with Intellectual Disabilities

**DOI:** 10.3390/bs15030281

**Published:** 2025-02-27

**Authors:** Sonia Hernández Hernández, Sergio Marín Quinto, Verónica Marina Guillén Martín, Cristina Mumbardó-Adam

**Affiliations:** 1Equity and Innovation in Education, Universidad de Cantabria, 39005 Santander, Spain; shh277@alumnos.unican.es; 2Department of Philology, Faculty of Philosophy and Letters, Universidad de Cádiz, 11519 Puerto Real, Spain; sergio.marinquinto@alum.uca.es; 3Deparment of Education, Universidad de Cantabria, 39005 Santander, Spain; 4Department of Cognition Development and Educational Psychology, University of Barcelona, 08193 Barcelona, Spain; cmumbardo@ub.edu

**Keywords:** pragmatics, communication, intellectual disability, assessment

## Abstract

People with intellectual disabilities live with significant conceptual, social, and practical limitations that hinder the acquisition, development, and use of language. Pragmatic skills facilitate interpersonal relationships, allowing for the understanding and expression of oneself, as well as the planning, organization, and adaptation of speech depending on the context and interlocutor. These skills imply, therefore, complex higher functions that must be articulated harmoniously for effective communication. Identifying the weaknesses of people with intellectual disability in the pragmatic dimension of language enables the provision of individualized support resources to guarantee their participation and social inclusion. This study presents a systematic review based on the PRISMA guidelines, and it includes the most commonly used assessment tools for pragmatic competence in people with intellectual disabilities over time. Of the 172 articles found, 20 met the inclusion criteria and were finally reviewed. The results show a lack of conformity between instruments in the pragmatic aspects evaluated and a lack of adjustment of the evaluation tools to the characteristics of this population. Therefore, the design of new standardized tests that specifically evaluate the pragmatic skills of people with intellectual disability is required in the near future. A tailored assessment is crucial for defining a complete profile of their communication skills and generating individualized intervention and support programs.

## 1. Introduction

In recent decades, neuropragmatics has become a central field of neuroscience research ([41]; [53]; [64]). This area of research stresses that people cannot account for communicative acts, both linguistic and non-linguistic ([60]), without referencing the underlying cognitive processes ([96]). Some authors even speak of cognitive pragmatics to refer to the analysis of the mental states of interlocutors ([7]). From this perspective, the existence of limitations in cognitive abilities or in the ability to discriminate the information implicit in a communicative exchange is likely to hinder the task of communicating ([8]), which may, in turn, affect both social skills and academic performance at an early age ([79]), as well as self-determination and socio-occupational inclusion at later developmental stages ([54]; [89]; [99]).

With this in mind, it is especially important to improve pragmatic competence within the field of intellectual disability (ID), which is currently defined, according to a socioecological approach, as a set of significant limitations both in intellectual functioning and adaptive behavior (expressed in conceptual, practical and social limitations), originating during the developmental period and restricting functional participation in everyday life ([82]). Although pragmatics has traditionally been valued as a strength of people with ID ([37]), individual differences in this population are highly manifested through different levels of support needs, making it necessary to develop specific evaluations in order to design personalized support plans that improve their social inclusion ([82]).

According to [2] ([2]), many people with ID live with communication barriers in their daily routines that limit their participation in both learning processes and social activities. This can be increased in people with specific disorders or syndromes, such as in the following: (1) people with autism and Fragile X syndrome (FXS), which is characterized by a limitation in initiating and maintaining conversations, as well as in literal comprehension resulting from a lack of attribution of intentions to the interlocutor ([24]); (2) people with Down syndrome (DS), who show a clear predominance of deficits in speech, coherence and use of context ([20]), with a lack of comprehension being a risk factor for the message being understood by the receiver ([51]); and (3) people with Williams syndrome (WS), whose difficulty in initiating a conversation appropriately or lack of control of proxemics ([42]) clearly hinders interaction.

According to [51] ([51]), it is essential to pay attention to language and communication in the field of disability, as failure in interpersonal relationships may lead to fewer future opportunities for access to employment ([31]). Behavioral problems might also arise from feelings of frustration ([87]), with the presence of emotional disorders being found in 91.3% of cases of young people with ID and associated language disorders ([9]). Consequently, research has shown that the presence of low language skills is related, among other things, to difficulties with social–emotional and behavioral adjustment ([16]; [28]; [47]; [90]; [97]; [105]), which may account for the negative attitudes that young people often display toward disability ([65]; [74]).

Based on the current ecological and biopsychosocial model of ID ([95]), priority areas for research and intervention must foster inclusion from a rights-based approach ([55]) and the improvement of their quality of life ([80], [81]; [94]) by adjusting the variables that guarantee universal and cognitive accessibility (Royal Legislative Decree 1/2013, 29 November). Therefore, it is essential to study the communication barriers that people with ID live with daily, and to emphasize the family, school, and socio-occupational environments as essential contexts for their social development and emotional regulation ([17]). Thus, from the perspective of quality of life and support ([95]), the services and resources provided must ensure compliance with the rights and well-being of people with disabilities. In addition, one of the main objectives of education must be to ensure that cognitive and/or communicative limitations do not represent a barrier or restriction to their right to full inclusion and participation ([103]). Likewise, flexibilization and adaptation measures should be facilitated to allow access to lifelong learning and increase their employability ([88]).

To make this possible, it is essential to have evaluative evidence with pragmatic foundations that “have an impact on the design of specific proposals and programs for communicative intervention” ([76]). Since [78] ([78]), the first to develop a global communication profile with The Functional Communication Profile, the increase in the design and development of resources for the evaluation of the pragmatic dimension aimed at different groups has been exponential. This increase has been noted both at the international level and in relation to the resources developed in Spanish ([70]). Although, in recent years, some previous tests have been revised and new tools have been published ([3]; [11]; [12]; [57]; [32]; [38]; [48]; [63]; [69]; [75]; [77]; [102]), these are not specifically designed and normalized for people with ID, who have a very heterogeneous linguistic profile ([14]). In addition, most of them do not account for the specificities of social communication across life stages ([70]), and they typically address the assessment of global communication from the approach of linguistic pragmatics ([35]), without focusing on specific pragmatic deficits that involve higher cognitive functions ([11]) or paying attention to “micro-skills”, which allow us to integrate the stimuli received to understand what others think ([33]). In addition, in general, these records are completed by the evaluator, and just a few consider the participation of significant people in the speaker’s life ([34]; [36]) or collect the information in natural or ecological interaction environments ([6]).

The aim of this research is to carry out a systematic review so as to obtain a global vision of the assessment of pragmatics in people with ID. Specifically, this study has four objectives: (O1) identifying studies where pragmatic language impairments have been specifically assessed in a population with ID, (O2) analyzing the profiles of those assessed, (O3) determining the most commonly used tools for this purpose, and (O4) examining the components of the pragmatic dimension on which research and complementary evaluations have focused. This information will have the potential to detect strengths and weaknesses that will guide the future of research in this field.

## 2. Materials and Methods

The systematic review was conducted following the PRISMA criteria (Figure 1) and according to the following process: (I) defining inclusion/exclusion criteria, (II) applying a search strategy, (III) selecting studies, (IV) data extraction, and (V) assessing studies’ risk of bias. PRISMA criteria can be found in Appendix A. The review was not prospectively registered. The whole process of the search and selection of studies is openly available in the Open Science Framework (OSF) (details in the Data Availability Statement). Each phase that was followed is described in a separate section below.

### 2.1. Inclusion Criteria

Studies that met the following inclusion criteria were included in the review: (I) All participants had to have ID, regardless of age, degree of impairment, or severity of symptoms. In studies with participants with different diagnoses, the results had to be reflected as categorized by disorder. (II) Studies focused on the assessment of pragmatic skills of subjects with ID (in the case of including the assessment of other skills, such as cognitive and/or linguistic skills, the results had to be shown separately). (III) Full texts were available and written in English or Spanish, independently of the participants’ nationality and the corresponding adaptation of the assessment tools to the language of the country where the research was conducted.

### 2.2. Search Strategy

The literature search for this systematic review was conducted on 29 February 2024 using the following databases: ProQuest, Scopus, PubMed and the core collection of Web of Science (WoS). These databases were selected because they are more closely linked to studies of a social and educational nature. The search equation was formulated using the following operators: (‘intellectual disabilit*’ OR ‘developmental disabilit*’ OR ‘learning disabilit*’ OR ‘mental retardation’) AND (‘pragmatic*’) AND (‘asses*’ OR ‘evaluat*’ OR ‘anali*’ OR ‘analy*’). The search process was not subject to a specific time interval due to the understanding of the scarcity of the existing literature. To ensure the quality and validity of the work, a peer review was performed, where the first and second authors shared parameters for analyzing the profiles of those assessed to help verify the results ([1]).

### 2.3. Selection of Studies

The first and second authors replicated the same process simultaneously and independently according to the established inclusion criteria and generated spaces for dialogue at the end of each phase. Initially, a filter was applied according to the titles and abstracts. Articles were imported into the bibliographic reference manager Mendeley Cite to automatically identify and eliminate duplicates, resulting in *n* = 172 studies. After an exhaustive analysis of the discrepancies and following the inclusion criteria, it was decided to include 6 texts that only one of the reviewers had originally selected. In addition, it was agreed to read the full texts of 17 articles that raised doubts.

Subsequently, after reading 22 full texts, the evaluators reached substantial agreement, with a coefficient of *K* = 0.72. Disagreement was detected in 10 texts, which, finally, were not included due to non-compliance with the inclusion criteria, resulting in total agreement and the definitive selection of 20 articles.

### 2.4. Data Extraction

The findings obtained in the assessment of pragmatic competence in people with ID were categorized according to the following variables: (I) article data (year, place of publication, country) and participant profile (number of subjects, gender, diagnosis and degree of impairment); (II) assessment variables measured (pragmatic tools, aspects assessed, other cognition and/or language tests, and collaborating informants), whose detailed analysis allowed for an overview of the current situation through the results and conclusions obtained in the different studies.

## 3. Results

Appendix A show the selected articles, and their objectives and designs. The most relevant results extracted from the 20 articles selected according to the different research questions are analyzed below (Table 1).

### 3.1. Analysis of the Studies About Pragmatics in ID (O1) and Profile of the Participants (O2)

As Table 1 indicates, the selected articles were published over the last three decades, with 2014 being the most prolific year, with 20% of the publications. In total, 50% of the research was carried out in European countries, while 35% was authored in the United States. The following were the types of studies: quasi-experimental studies (50%), phenomenological studies (20%), case studies (10%), psychometric property studies (10%), systematic reviews (5%), and case series studies (5%). Considering the three levels of pragmatics ([32]); (enunciative (i.e., locutive and illocutive dimensions), textual (i.e., word and sentence construction, argumentative superstructures), and interactive (e.g., turn-taking)), our findings reveal that the majority of the studies were based on interactional pragmatics (60%). In contrast, only 25% of the research publications aimed to assess enunciative or textual pragmatics. The remaining 15% of the articles were just dedicated to testing the validity of a tool or intervention.

The total number of subjects with ID evaluated was 650, and their profiles are classified as follows: people with idiopathic ID (35.2%); Down syndrome (DS) (16.05%); Williams syndrome (WS) (13.4%); Fragile X syndrome (FXS) (13%); FXS and/or people with autism (19%). As for the severity of the ID, subjects presented as follows: mild (20%), mild–moderate (35%), moderate (15%), mild–deep (5%), and not specified (25%). Only 30% of the papers were based on specific studies of a single group (20% on WS and 10% on DS), while 11% were comparative studies. The only two case studies focused on people with autism. Age was only specified in 75% of the studies, with range being between 20 months and 40 years. Adolescence was the most represented range. In relation to sex, this was detailed in 60% of the studies, with a computation of 92 males and 90 females.

### 3.2. Pragmatics Tools Used to Assess ID Population (O3)

Regarding the assessment tools used in the different selected studies (Table 2), two categories can be established: (1) those using standardized tests, interviews, scales, or checklists carried out in structured situations (90%) and (2) those using direct observation in spontaneous or semi-structured situations (10%). Thirty percent of the studies used both modalities. On the other hand, 60% of the studies applied an assessment tool to a person with an ID, while 15% applied it to an informant, and 25% used both modalities.

#### 3.2.1. Scales and Checklists

Appendix A summarizes the evaluation instruments of the selected studies.

##### Most Commonly Used Tools

(1) The Vineland Adaptive Behavior Scales (VABS) ([86]) incorporate a semi-structured parent interview to measure adaptive behavior. Although this scale was not specifically developed to measure pragmatic language, it allows information to be collected in different domains related to communication, daily living skills and socialization. [4] ([4]) revealed that the WS child population obtained data similar to those of their peers in socio-communicative skills. The results were even above the norm in children and adolescents. On the other hand, they revealed that the communicative resources of people with ID are functional. They showed that communicative behavior is affected when the context in which a child is assessed is changed ([44]) and that improvements in adaptive behavior are sometimes observed ([18]).

(2) Children’s Communication Checklist-2 (CCC-2) ([11]) is a questionnaire designed to measure various aspects of communication covering different language skills. It can be administered to parents and teachers of children aged 4–16 years to collect data on all or most of the following subscales: speech, syntax, semantics, coherence, inappropriate initiation, stereotyped language, use of context, non-verbal communication, social relationships, and interests. In [100] ([100]), 39 of the 70 items were assessed and found to reflect a greater deficit in the pragmatic skills of subjects with ID compared with the norm. On the other hand, [43] ([43]) and [21] ([21]) complemented the results obtained by analyzing specific etiologies and/or administrating additional tests, with the former detecting that there were no significant differences between the CCC-2 ([11]) and the TOPL-2 ([69]) scores, while the latter reported that participants with DS demonstrated greater competence in pragmatic skills than participants with FXS before controlling for non-verbal cognition.

##### Less Frequently Used Tools

(1) Comprehensive Assessment of Spoken Language—Pragmatic Judgement (CASL- PJ) ([15]) was used by [49] ([49]) and [58] ([58]); it involves telling a short story to subjects (aged 3–21) about children in different social situations. It assesses the ability to provide a pragmatically appropriate response about what they should do or say while considering communicative intention, turn-taking, emotional expression, and pragmatic appropriateness. In both studies, they showed that lower scores were closely related to autism comorbidity.

(2) The Turkish Version of the Pragmatic Language Skills Inventory (TV-PLSI) is a Turkish version of the previous item. It is a normative assessment tool for children aged 5–12 years, consisting of 45 items divided into three subscales: (1) stereotypical behaviors, (2) communication and social interaction, and (3) classroom, social and personal interaction skills. A rating scale is used and administrated by the primary caregiver or teacher to the individual in 5 or 10 min. [25] ([25]) set out to compare the pragmatic skills of children with Autism and ID through teacher reports, finding them to be below average.

(3) [44] ([44]), in addition to implementing the VABS, added the Macarthur Communicative Developmental Inventory (MCDI) ([30]), an instrument completed by parents or professionals and designed for children between 8 and 30 months. It assesses early language acquisition through prelinguistic indicators, vocabulary, and grammar. It does so with the aim of comparing structured versus unstructured conditions, and it was concluded that structured procedures (communicative temptations) are an effective and efficient means of assessing early communicative skills.

(4) Pragmatic profile of early communication skills ([22]): [45] ([45]) compared the pre-linguistic skills of children with DS with those of children with TD and with different learning difficulties. They used this questionnaire, which is completed by parents and assesses infants’ functional communication skills. They found that children with DS and infants with TD did not differ significantly in some non-verbal communication behaviors.

(5) In Systematic Analysis of Language Transcripts (SALT), [21] ([21]) incorporated aspects such as physical context, audience, topic, purpose, visual and gestural cues, and abstractions. The results point to better competencies in people with DS than in people with FXS.

(6) Test of Pragmatic Language—2 (TOPL-2) ([69]). Here, [43] ([43]) used the TOPL-2 to complete an assessment of the CCC-2. It includes 43 items for individuals aged 8 to 18 years and has a 17-item version for children aged 6 to 7 years. It is divided into seven pragmatic areas—physical context, audience, topic, purpose, visuospatial cues, abstractions, and pragmatic evaluation, while also allowing other variables, such as loquacity, unintelligibility, disfluency, lexical diversity, syntactic complexity, inappropriate initiation, stereotyped language, use of context, and non-verbal communication.

(7) Pragmatic Rating Scale-School Age (PRS-SA) ([39]; [52]) assesses 34 characteristics of pragmatic behavior during a semi-structured social interaction, such as verbosity, social opportunity, thematic continuity, redundancy, failure to initiate a sentence, reduced communicative intention, inappropriate turn-taking, and the use of non-verbal behaviors such as eye contact and communicative gestures. In addition, the items are scored on a scale of 0–2, meaning that a higher score implies a greater pragmatic deficit. The results of [49] ([49]) showed that children with autism and FXS + autism presented problems in mood signaling and in paralinguistic intelligibility and stuttering aspects, while children with DS exhibited difficulties in stuttering, cluttering and intelligibility.

(8) [92] ([92]) used the Subtest Formulating Sentences of the Dutch adaptation of Clinical Assessment of Language Fundamentals- Fourth Edition (CELF-4-NL) ([50]) and identified that children with 22q11.2DS, with ID, and with ID + autism make errors in the form, content, or use of language due to incorrect contextual interpretation.

(9) [91] ([91]) organized the pragmatic skills of children with WS and ID into four categories: (1) perspective-taking ability, as assessed with a Dutch adaptation of the Action Picture Test ([72]); (2) quality and quantity of information transferred, with the Action Picture Test—Information Transfer Score (ITS-APT); (3) manner and relevance of information reporting, as assessed by coding sentences into five main themes; (4) narrative ability, as assessed using the Information Transfer Scores—Bus Story Test (ITS-BST) ([73]). As a result, children with WS exhibited a lower mean score for adequate information transfer and retelling compared with children with ID.

Finally, [26] ([26]) conducted a systematic review on the assessment of paralinguistic aspects in people with ID, such as emotion recognition. The results revealed that nonverbal speech information has been frequently assessed in the syndromic ID population by means of tools such as the following: Diagnostic Analysis of Nonverbal Accuracy Scale-2 ([66]), Florida Affect Battery ([13]), Test of Facial Emotion Recognition ([104]), and Self-Report Anger Inventory ([10]). This systematic review is limited to analyzing aspects of comprehension and expression of paralinguistic elements of communication in adults with ID, and does not show results about linguistic pragmatics.

##### Studies in Which a Pragmatic Checklist Is Designed

(1) [40] ([40]) designed a pragmatic checklist where, by means of a semi-structured interview and observation, they determined whether there were changes in a subject’s communicative skills after the implementation of a speech-generating device. They conducted an analysis of the communicative exchange through an ad hoc designed checklist composed of 15 pragmatic dimensions: (1) initiation of the interaction, (2) introduction of a specific topic, (3) continuation of the communication topic, (4) respect of the conversational turn, (5) taking of the conversational turn, (6) maintenance of the topic after several turns of communication, (7) communication interruption, (8) request for clarification from the conversational partner, (9) restoration of the communication interruption, (10) completion of the conversational topic, (11) introduction of new conversational topics by the subject, (12) introduction of a new topic by the conversational partner, (13) maintenance of appropriate conversational eye contact, (14) use of gestures, and (15) use of facial expression. As a result, they found that a participant with moderate ID improved in turn-taking, topic maintenance, and the incorporation of new topics with support.

(2) [59] ([59]) designed the Contextual Assessment Inventory, where a contextual assessment is conducted, and this appears to be efficient, thorough, and comprehensive, but does not provide information on why a particular item is associated with higher rates of problematic behavior.

#### 3.2.2. Direct Observation

Direct observation (OD) was used as a tool for analyzing pragmatics in a total of eight studies. Those that applied it as a single test were the following: (1) [5] ([5]) for the systematic training of people with DS, observing improvements in turn-taking, asking questions, improving eye contact, and tone of voice; (2) [67] ([67]), who concluded that the most commonly used speech act among people with ID who were users of a residence was declarative, using discussion instead of dialogue.

Other studies using OD to complement their assessment include the following: (1) [23] ([23]) assessed the coherence and cohesion of the discourse of young adults with WS; (2) [40] ([40]) designed a pragmatic checklist analyzing the 15 variables of the communicative behavior of a person with ID from the information that they collected through direct observation in their residential environment; (3) [44] ([44]) evaluated communicative acts through unstructured play with a child; (4) [45] ([45]) evaluated pre-linguistic skills through an unstructured play situation; (5) [46] ([46]) evaluated verbal extension in a conversational exchange with a group of 14 children with WS based on play with their mothers; (6) [59] ([59]) designed a test (Contextual Assessment Inventory) for non-standardized assessment by collecting paralinguistic, contextual, or communicative interaction aspects (turns, speech acts, initiatives, etc.). They concluded that this information can help predict greater markers of difficulty in pragmatics.

As part of the evaluation process, another significant variable considered in 55% of the articles was the figure of an informant, who was usually represented either by the family (63.6%) or by professionals (36.4%). As far as the family was concerned, in most cases, either the mother or both parents were usually involved (assigning each one a different task). The informant’s role was to collaborate in data collection by providing additional information through different formats, such as interviews, filling in scales, questionnaires, or checklists, or being part of a spontaneous play situation where the mother interacted with the subject–participant of the study. Regarding professionals, 33% of these collaborators were support staff at a center (teachers, carers, etc.).

### 3.3. Components of the Pragmatic Dimension and Other Assessed Aspects (O4)

The domains or components of the pragmatic dimension analyzed were the following: (1) behavior, which was analyzed in terms of adaptive behavior and stereotyped behavior; (2) daily living skills related to habitual tasks and routines; (3) social interaction skills, which were focused on the pragmatic use of language appropriate to the social context; (4) interpretation of contextual cues considering the environment and flexibility to change from one context to another; (5) conversational skills based on turn-taking and turn respect, thematic continuity, failure to introduce new topics, conversational initiative, speech acts, and communicative intention; (6) narrative ability, considering the quality and quantity of information transfer, vocabulary, grammatical and lexical structure of stories, the redundancy of topics, and the coherence and cohesion of narrative structure; (7) non-verbal communication, assessed in terms of eye contact, use of gestures and facial expressions, and the recognition of emotional prosody and emotions articulated through facial expressions; (8) requests for clarification; (9) pre-linguistic skills. The types of natural environments selected for direct observation were usually the following (from the highest to the lowest prevalence): home, hospital, educational or work center, and, in only one case, the subjects’ immediate community.

Finally, only 15% of the studies exclusively reviewed the pragmatic dimension. In people with ID, 85% of the studies (*n* = 14), in addition to using pragmatic competence assessment instruments, added other complementary tests that assessed cognition and language. Of the total number of assessment tools, 28.5% used only one test, mostly related to language. Studies where complementary tests were not applied were observational. The most frequently used cognition tests were the following: Leiter International Performance Scale-Revised (Leiter-R), ([29]); Wechsler Preschool and Primary Scale of Intelligence-III, Dutch adaptation (WPPSI-III-NL), ([98]). The most commonly used language tests were Peabody Picture Vocabulary Test (PPTV), ([27]) and the Dutch version of the Receptive Vocabulary Age Equivalents (RVAE) (PPVT-III-NL), ([83]); Clinical Evaluation of Language Fundamentals ([84]; [102])—including the Preschool Edition (CELF-P2, [101] and the Dutch adaptations (CELF–4–NL, [50]; CELF–P2–NL, [19]); and Comprehensive Assessment of Spoken Language (CASL)—Antonyms ([71]).

### 3.4. Risk of Bias

Finally, the risk of bias of the 20 articles was analyzed by classifying them by the type of study based on the domains of the Joanna Briggs Institute ([62]).

(1) Qualitative phenomenological studies: The results were favorable with respect to inconsistency (philosophical perspective, objectives, and data collection), as the studies were consistent with the research methodology, objectives, and data collection. [46] ([46]) used transcription and coding to assess the sentences uttered by the participants; however, the results were not analyzed with a phenomenological approach, and standardized measures were used without considering individual differences. The aim of [85] ([85]) was to find out, through a qualitative methodology, whether there were differences in people with mild ID according to their gender or age. However, the study is flawed in terms of its methodology, as it does not attempt to delve into the phenomenology of the study subjects but rather seeks to generalize the data to develop standardized scores and, consequently, to develop an intervention guide for improving pragmatic skills in academic and social contexts. Another design should have been employed rather than a descriptive–analytic and causal non-experimental methodology, as it also defines the dependent (pragmatic skills) and independent (gender and age) variables of the study. It is also noteworthy that, except for that of [67] ([67]), none of the studies clearly stated the researchers’ cultural or theoretical stances on the research being conducted. In general, the representation of the participants, moral approval by parents and an ethical committee, and the conclusions drawn are criteria that are met in all of the selected articles; therefore, they present a low risk of bias.

(2) Case studies: The two case studies ([18]; [40]) presented an elaborate methodological design, resulting in a low risk of bias in any of the aspects assessed.

(3) Case series studies: In the study by [5] ([5]), there was a high risk of bias in most aspects, with the exception of the demographic characteristics and clinical history of the participants, the presentation of the results, and the demographic information of the clinical setting, as these four sections were well defined in the article.

(4) Quasi-experimental studies: As these were studies assessing pragmatic competence, the identification of two aspects related to the experimental design as subjecting participants to an intervention and pre- and post-intervention measures was impossible. None of the studies analyzed developed an intervention; therefore, no pre- or post-intervention measures were taken. However, the existence of the control group was determined by the design of the study itself in some studies ([45]; [49]; [58]; [91], [92]) in which a specific syndrome was addressed, and the purpose was to identify pragmatic skills in comparison with those exhibited by other subjects with different syndromes that also cause ID. In these studies, all participants were assessed with the same instrument. However, it should be noted that in the study by [91] ([91]), in the second assessment, which was performed five years after the first measurement, each child had a different profile (regardless of which group they belonged to—WS or idiopathic ID); while some had attended speech therapy, others had not; some were in a special education school, while others were in mainstream schools. This could have biased the differences in the results obtained in the second measurement. Other studies ([4]; [21]; [25]; [43]; [44]) aimed to determine the pragmatic skills of subjects with different diagnoses or with the same diagnosis but at different ages. Their purpose was not to study the characteristics of a single syndrome in depth, but to draw up a general profile of the pragmatic skills of subjects with different syndromes or to find out the evolution of pragmatic skills at different ages. For this reason, there was no control group, since all of the participants made up the group of interest, but there were groups that were differentiated according to diagnosis and/or age. Four of the studies ([4]; [21]; [25]; [43]) did not accurately build the reliability of their results. In the case of [21] ([21]), the study indicated that all measurements were administered by well-trained assistants with extensive experience working with people with developmental disorders; however, they did not indicate the reliability of the scores obtained, nor the degree of inter-rater agreement. Similarly, Hoffmann et al. (2013) specified that the implementation of the assessment instrument was performed by a speech therapist, but did not provide the reliability and validity of the scores. Meanwhile, the studies by [4] ([4]) and [25] ([25]) indicated that all the instruments used had a high degree of validity and reliability, but they did not provide the reliability and validity of the scores obtained in their studies. However, even though the reliability of the results is not clear in all of the articles, none of the studies presented a high risk of bias in any of the other aspects assessed, which lends credibility to their assessment of pragmatic skills in subjects with ID.

(5) Studies of psychometric properties: With regard to the psychometric properties in the studies, it should be noted that the only two points of concern were the criterion validity, since no study provided information about this, and reliability, since in the study by [59] ([59]), the reliability coefficient was <0.70, and in that by [100] ([100]), they again did not provide the data. The rest of the domains assessed were well elaborated in the studies; therefore, they presented a low risk of bias.

(6) Systematic review: The review by [26] ([26]) fulfilled only two of the eleven established criteria, presenting a high risk of bias in the absence of two reviewers for critical appraisal, in the methodology used to minimize data extraction and combine studies, and in the non-existent assessment of bias in the publications. Overall, the studies have a low risk of bias, considering that the design, evidence, and results are valid.

## 4. Discussion

Neuropragmatics has emerged as a crucial field in the understanding of communicative acts, highlighting the interrelationship between cognitive processes and communication. When elaborated language levels are not available, primitive forms are sometimes used to express oneself ([93]) and, although maladaptive behaviors (avoidance, shutdown, aggression, disruption, etc.) have always been considered to be inherent to ID, they are now recognized as a manifestation of the consequences resulting from failure in interpersonal relationships or barriers to participation, thus conditioning their self-determination and empowerment, among other aspects ([82]).

After compiling the available studies on the assessment of pragmatics in people with ID, a significant number of texts were not found, despite not having narrowed the time range of the search and not having established very restrictive inclusion criteria. Furthermore, there has been a lack of publications in recent years, with the interval of greatest scientific dissemination being between 2013 and 2018. In short, research on the pragmatic skills of people with ID can be considered relatively novel, and is still at a very early stage. Evidence suggests that, with respect to a certain period (1950–2000), the number of tools developed and/or revised to assess pragmatic competence has increased exponentially in this century ([48]; [102]). Although there is unanimity on the purpose of the 20 texts collected in this study, with pragmatics being, in general terms, their objective, there is a great deal of variability in the constructs assessed, the instruments used for this purpose, and even in the components of pragmatics that are considered most significant for determining the communicative profile of the subjects. There are hardly any two identical evaluations that repeat the structure or elements of the process, even when assessing the same population, such as people with DS ([45]; [67]), or when evaluations are executed by the same evaluator, as in the case of [92] ([91], [92]).

In short, there is considerable heterogeneity in the selection of tools used to assess pragmatic competence. In general, observation is the most widely used tool, given the difficulties in standardizing scales. Among these, the most widely used tools are the Vineland Adaptive Behavior Scales (VABS) ([4]; [18]; [44]), which record the domains of adaptive behavior, communication, daily living skills and socialization, and categorize these as speech acts, and the Children’s Communication Checklist-2 (CCC-2) ([21]; [43]; [100]), whose approach fits more closely with the assessment of linguistic pragmatics. Although both scales are standardized scales, none of them have been specifically designed and normed for evaluating communication in people with ID. For that, direct observation is the methodological resource that is most commonly used in these cases, both exclusively or in addition to other instruments ([45]; [46]; [59]), where the common denominator is contextual pragmatics or interaction structure.

As for the use of other tests applied as a complement to pragmatic language assessments, only 11 studies consider cognitive assessment to be relevant ([4]; [18]; [21]; [25]; [26]; [43]; [44]; [46]; [49]; [58]; [91], [92]). More studies focus on assessing subjects’ linguistic repertoire, that is, the content and form of language ([4]; [25]; [26]; [43]; [44]; [45]; [46]; [49]; [58]; [91], [92]). Only one of the studies ([100]) completed a pragmatic assessment test with several subscales measuring social behavior and aspects of context.

In terms of methodology, more than half of the studies considered the participation of informants, with the aim of collaborating in data collection, providing additional information, and/or being part of the spontaneous situation of semi-structured or spontaneous conversation. These figures mainly represent the parents (both parents or only the mother), professionals, and/or carers.

In relation to the analyses of subjects with ID under evaluation, it is important to note that there is parity between sexes, the average age is in the adolescent stage, and the studies are based on the same etiologies; the diagnoses of the participants are limited to idiopathic ID, WS, DS, XFS, and/or autism. No studies on low-prevalence neurodevelopmental syndromes or disorders are included, except in two cases—[92] ([92]) and [100] ([100]), who include, in addition to other more prevalent groups, 18 subjects with 22q11.DS and 144 people with “different mental conditions”, respectively. In conclusion, it seems that there is a certain homogeneity in the participants recruited for the different studies, suggesting that other profiles may be overlooked in this area of evaluation.

People with WS are the group that has most often been the subject of pragmatics studies for several purposes, as follows: (1) delimitating a behavioral phenotype related to some pragmatic skills (retelling a story and referential communication) and assessing structural language skills in the longitudinal study conducted by [92] ([92]); (2) determining narrative coherence and cohesion ([23]); (3) comparing two pragmatic assessment tools and concluding on whether they are valid in identifying the pragmatic difficulties of this group ([43]); (4) establishing the temporal stability of different pragmatic skills (number of sentences and questions a child asks to lengthen a conversation) in a longitudinal study ([46]); (5) determining whether social–communicative skills are in line with mental age by comparing two age groups: children and adolescents ([4]). The next groups in which a higher frequency of pragmatics assessment has been found are people with DS and SXF ([21]; [45]; [49]; [58]).

Some studies have compared pragmatic skills between different syndromes ([21]; [49]; [58]), and others compared pragmatic skills with linguistic skills or with autism to determine their influence on pragmatic difficulties ([4]; [25]; [91], [92]). Regarding these studies’ results, there are some false beliefs that have been refuted, such as that people with WS are “natural story tellers” ([23]), supporting other studies wherein people with WS produced narratives that lacked coherence, and many times obtained worse scores in pragmatic skills than people with DS ([61]). Similarly, it seems that there are not so many differences in the pre-linguistic skills of children with DS and TD ([45]), as was found in other research.

This systematic review can be considered reliable, as the entire procedure was conducted using a peer-review approach. Each researcher replicated the entire process independently, obtaining concordant results. As for the limitations of the study, given the variability of meanings used to refer to the construct of pragmatics, the assessment tools and papers reviewed are very heterogenous and difficult to compare. In addition, it is possible that the search expression or the eligibility criteria established may have resulted in potentially important information being discarded. Lastly, studies of subjects with Autism or ADHD profiles that did not present ID were eliminated; however, it would be important to take into consideration certain variables of neurocognition that limit the functioning of the communication process, such as theory of mind, central coherence theory, and executive function disorder, a triad on which there is already a large body of literature that could serve as a basis for identifying other important assessment models.

Future research must include a more representative sample of people with ID, striving to encompass a more diverse sample. It is also necessary to design a pragmatic assessment scale for people with ID that considers their specificity and allows a complete profile of their communicative potential and weaknesses to be defined. Consequently, professionals could more easily develop individualized support plans that would meet the needs of each person in their natural environment, thereby eliminating social stigma and empowering people with ID. Similarly, pragmatic assessment tools for people with ID must consider different age groups and should provide shared patterns among people with the same profile. In order for this new tool to be as comprehensive and reliable as possible, it would be important to incorporate variables that are barely observed in the selected studies, or that would need to be studied in greater depth, such as the following:

(I) Augmentative and/or alternative communication, considering that a very high percentage of people with ID are likely to be users of a communication support product due to the significant limitations that characterize the group. Only one of the studies mentions comparing the performance of a subject after training with a speech-generating instrument ([40]). As development progresses, problems decrease or disappear ([56]);

(II) A detailed analysis of the context wherein the subject operates, in order to know the external demand and assess the adaptive skills necessary to adjust to both the environment and the interlocutors (cognitive accessibility, opportunities, communication styles, etc.);

(III) The prerequisites of cognition, such as attention (assessing, among other things, the time of sustained attention during a communicative exchange) or aspects related to neurocognition (decision-making based on reasoning or the integration of information that occurs during interaction);

(IV) The participation of various close people as main informants as a complement to an expert’s assessment. They become allies for therapists in terms of both the collection of data and the generalization of learning, where it is understood that, despite the risk of bias due to subjectivity or logical ignorance, they are natural supports, they know the subject, and they can provide reliable information on their level of performance in various real situations.

## Figures and Tables

**Figure 1 behavsci-15-00281-f001:**
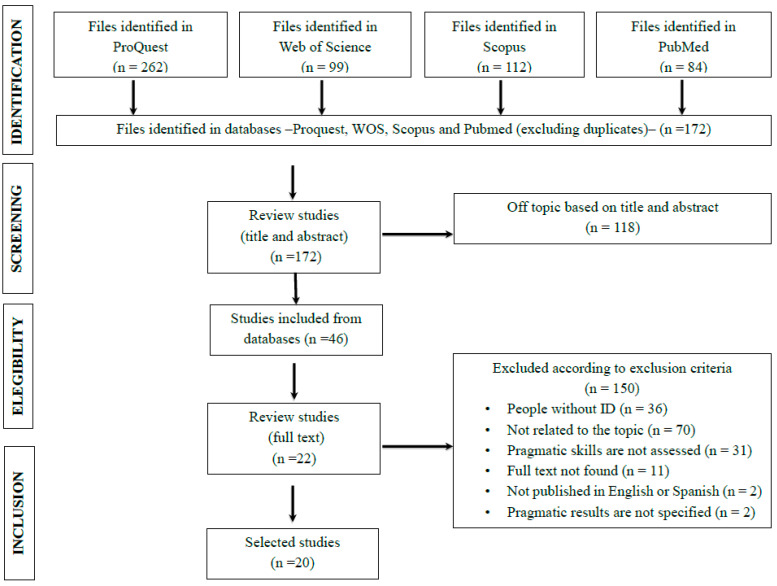
Flowchart (PRISMA) of the study selection procedure.

**Table 1 behavsci-15-00281-t001:** Characteristics of the participants of the selected studies.

Selected Articles		Country	N	Diagnosis	Age	Sex
1	[4] ([4]).	Italy	32	WS	Children (14.6 years)/adolescents/adults (39.8 years)	14 girls/18 boys
2	[5] ([5]).	USA	5	(1) DS; (2) Idiopathic ID; (3) DI due to trauma	15–19 years	4 men/1 woman
3	[18] ([18]).	Canada	1	Autism + ID	Adolescent	Male
4	[21] ([21]).	USA	69	(1) 39 FXS; (2) 30 DS		(1) FXS (men); (2) DS (20 men/10 women)
5	[23] ([23]).	Spain	9	WS		5 women/4 men
6	[25] ([25]).	Turkey	86	(1) 34 Autism; (2) 52 ID		Autism (24 boys/10 girls)
7	[26] ([26]).	Serbia		(1) WS; (2) Idiopathic ID; (3) DS	>17 years	
8	[40] ([40]).	Ireland	1	Autism + ID		Woman
9	[43] ([43]).	USA	20	WS	6–16 years	
10	[44] ([44]).	Australia	11	(1) DS; (2) TND		7 men/4 women
11	[45] ([45]).	UK	10	SD	21–53 months	6 boys/4 girls
12	[46] ([46]).	USA	14	WS	boys 4.30 years/girls 10.18 years	7 boys/7 girls
13	[49] ([49]).	USA	123	(1) 29 Idiopathic Autism; (2) 38 FXS + comorbid; (3) 16 FXS without Autism; (4) 20 DS; (5) 20 TD	(1) Idiopathic Autism (school-age children); (2) FXS + comorbid; (3) FXS without Autism (not specified); (4) DS 11.31 years; (5) TD 4.82 years	
14	[58] ([58]).	USA	151	(1) 29 FXS; (2) 40 FXS + Autism; (3) 34 DS; (4) 48 TD		
15	[59] ([59]).	USA	20	(1) Autism + ID; (2) FCU+ ID; (3) TB + ID; (4) Epilepsy + ID; (5) DT + ID		13 men/7 women
16	[67] ([67]).	Canada	4	DS	40, 30, 31, 32 years	Men
17	[85] ([85]).	Slovenia	60	Mild ID	30 students (7 years old—15 girls and 15 boys)/30 students (9 years old—15 girls and 15 boys)	
18	[91] ([91]).	Belgium	24	(1) 12 WS; (2) 12 Idiopathic ID		
19	[92] ([92]).	Belgium	90	(1) 18 with 22q11.DS; (2) 19 Idiopathic ID; (3) 23 Idiopathic ID + Autism; (4) 30 DT		53 men/37 women
20	[100] ([100]).	Germany	839	(1) 195 Autism; (2) 83 ID; (3) 144 different mental conditions; (4) 417 DT		

ID, intellectual disability; FXS, Fragile X syndrome; DS, Down syndrome; WS, Williams syndrome; TND, neurodevelopmental disorder; DT, typical development; TB, Bipolar disorder; PKU, phenylketonuria; TD, depressive disorder.

**Table 2 behavsci-15-00281-t002:** Evaluation instruments of the selected studies.

Selected Articles	Pragmatic Evaluation Tools	Components of Pragmatics	Other Cognitive or Language Tests	Informant
[4] ([4]).	VABS	VABS Communication Mastery	Leiter-R, PPVT-r, PVCL, BNT	Parents
[5] ([5]).	OD	(1) Mandatory turn-taking (2) Non-mandatory turn-taking (3) Ask focused questions to one’s partner (4) Conversing with appropriate eye contact (5) Using an appropriate tone of voice in conversational speech		
[18] ([18]).	VABS-II	(1) Adaptive behaviors (2) Communication (3) Daily living skills (4) Socialization	RPM, WISC-V, CELF-CDN-F, WIAT-II, EOWPVT-IV, EVIP, LE Vol du PC, BALE	1 (Mother)
[21] ([21]).	(1) CCC-2 (2) SALT	(1) Loquacity (2) Unintelligibility (3) Disfluency (4) Lexical diversity (5) Syntactic complexity (6) Inappropriate initiation (7) Stereotyped language (8) Use of context (9) Non-verbal communication	Leiter-R, PPVT-III, CASL-EVT, CASL-SC, FBT, CBCL/6–18	2 (Parents)
[23] ([23]).	(1) OD (2) Recount of the sequential order of events. (3) Block design. (4) Average use of discourse markers (5) “Frog Goes to Dinner”	Coherence and cohesion of pragmatic competence	PPVT, WAIS-III	
[25] ([25]).	TV-PLSI	(1) Stereotypical behaviors, communication and social interaction (2) Classroom interaction skills, social and personal interaction skills	TV-GARS-2	Special education school teachers
[26] ([26]).	(1) Electrophysiological study (2) DANVA-2 (3) FAB (4) TFER (5) SRAI (6) Photos with facial expression	Recognizing emotional prosody and emotions	(1) Intelligence test (2) Visual discrimination	
[40] ([40]).	(1) Pragmatic checklist designed by researchers (2) OD	(1) Beginning of communicative interaction. (2) Presentation of a specific topic. (3) Respect for taking turns. (4) Thematic continuity. (5) Request for clarification. (6) Appropriate eye contact. (7) Gestures and facial expressions.		One day center member and one residential center person
[43] ([43]).	(1) TOPL-2 (2) CCC-2	(1) TOPL-2: physical context, audience, theme, purpose, visual and gestural cues, abstractions and pragmatic evaluation (2) CCC-2: pragmatics, syntax, morphology, semantics and speech	KBIT-2	Parents
[44] ([44]).	(1) VABS (2) MCDI (3) OD	(1) Requests and comments. (2) Communicative acts.	RDLS-R	1 (Mother)
[45] ([45]).	(1) “Pragmatic profile of early communication skills” (2) OD	Pre-linguistic skills.	DLS	(1) Father (2) Mother
[46] ([46]).	(1) OD based on the game (2) Conversation with a family researcher.	(1) Number of sentences that the child makes and serves to adapt to the context (2) Questions in which the child makes eye contact with the interlocutor (3) Eye contact that the child makes while producing the statements	DAS, EVT	Mothers
[49] ([49]).	(1) CASL–PJ (2) PRS–SA	(1) Communicative intention, turn-taking, emotional expression and pragmatic adaptation (2) Verbosity, social appropriateness, scripting, redundancy, failure to initiate topics, inappropriate turns, eye contact and communicative gestures	Leiter-R, PPVT-III, EVT, ADOS	
[58] ([58]).	CASL-PJ	Knowledge of appropriate language for various social situations	Leiter-R, CASL-Antonyms, CASL-SC, ADOS	
[59] ([59]).	(1) Checklist CAI (2) OD	(1) Negative interactions and disagreements (2) Factors related to tasks and factors related to daily routines. (3) Uncomfortable environment and changes in the environment		2 (Support staff)
[67] ([67]).	OD	Speech acts, communicative initiative, speaking turns, and types of statements		1 man/1 woman (Support staff)
[85] ([85]).	The Storytelling Test: Illustrations of The Frog King	Vocabulary, grammatical structure and structure of the content of stories		
[91] ([91]).	(1) APT (2) ITS-APT (3) RTNA-BST (4) Coding of statements into five categories (5) ITS-BST	(1) Perspective-taking ability (2) Quality and quantity of information transfer. (3) Manner and relevance of information transfer: coding of the statements into five categories (4) Narrative capacity	WPPSI-III-NL, SON, PPVT-III-NL, CELF-P2-NL, CELF-4-NL	
[92] ([92]).	FS (CELF-5)	Ability to interpret and use contextual information	WPPSI-III-NL, SON R6–40, PPVT-III-NL, CELF-4-NL, CELF-P2-NL, RS	
[100] ([100]).	CCC-R, CCC-2	This is made up of 39 items instead of the 70 items that made up the original checklist	CBCL, SRS, SQC, Demographics data sheet	

Abbreviations of pragmatic evaluation tools. VABS: Vineland Adaptive Behavior Scales. FS: Subtest Formulating Sentences Clinical Assessment of Language Fundamentals-Fifth Edition (CELF-5). CCC-R: Revised Children’s Communication Checklist-2. CCC-2: Children’s Communication Checklist-2. CASL-PJ: Comprehensive Assessment of Spoken Language—Pragmatic Judgement. PRS–SA: Pragmatic Rating Scale—School Age. ADOS: Autism Diagnostic Observational Schedule. SALT: Systematic Analysis of Language Transcripts. APT: Action Picture Test. ITS-APT: Information Transfer Score. RTNA-BST: Renfrew Language Scales Dutch Adaptation—Bus Story Test. ITS-BST: “Information Transfer Score”. MCDI: Macarthur Communicative Developmental Inventory. TV-PLSI: Turkish Version of the Pragmatic Language Skills Inventory, OD: Observación Directa. CAI: Contextual Assessment Inventory. TOPL-2: Test of Pragmatic Language-2. DANVA-2: Diagnostic Analysis of Nonverbal Accurcy Scale-2. FAB: Florida Affect Battery. TFER: Test of Facial Emotion Recognition. SRAI: Self-Report Anger Inventory. Abbreviations of other cognitive and language test. RPM: Raven’s Standard Progressive Matrices. WISC-V: Wechsler Intelligence Scales for Children—Fifth Edition. CELF-CDN-F: Clinical Evaluation of Language Fundamentals—French Canadian Version. WIAT-II: Wechsler Individual Achievement Test—Second Edition. EOWPVT-IV: Expressive One-Word Picture Vocabulary Test—Fourth Edition. EVIP: Échelle de Vocabulaire en Images Peabody (French Version of the Peabody Picture Vocabulary Test). BALE: Batterie Analytique du Langage Écrit. WPPSI–III–NL: Wechsler Preschool and Primary Scale of Intelligence–III, Dutch Edition. SON R6–40: Categories and Analogies subtests from the Snijders–Oomen Nonverbal Intelligence Test Revised age 6–40. PPVT–III–NL: Receptive Vocabulary Age Equivalents (RVAE) of the Dutch edition of the Peabody Picture Vocabulary Test. CELF–4–NL: Dutch adaptation of the Clinical Evaluation of Language Fundamentals–Fourth Edition. CELF-P2-NL: Clinical Evaluation of Language Fundamentals—Preschool–Second Edition. RS: The Recalling Sentences. CBCL: Child Behavior Checklist. SRS: Social Responsiveness Scale. SCQ: Social Communication Questionnaire. Leiter-R: Leiter International Performance Scale-Revised. ADOS: Autism Diagnostic Observation Schedule. CASL-Antonyms: Comprehensive Assessment of Spoken Language—Antonyms. CASL-SC: Comprehensive Assessment of Spoken Language—Syntax Construction. PPVT-III: Peabody Picture Vocabulary Test—Third Edition. CASL-EVP: Comprehensive Assessment of Spoken Language—Expressive Vocabulary Test. FBT: False Belief Task. CBCL/6–18: Child Behavior Checklist, Ages 6–18. WAIS-III: Wechsler Adult Intelligence Scale. RDLS-R: Reynell Developmental Language Scales—Revised. TV-GARS-2: Turkish Version of the Gilliam Autism Rating Scale-2. DLS: Derbyshire Language Scheme. KBIT-2: The Kaufman Brief Intelligence Test, Second Edition. DAS: Differential Ability Scales. PPVT-r: Peabody Picture Vocabulary Test—Revised. PVCL: Prove di Valutazione delle Competenze Linguistiche. BNT: Boston Naming Test.

## Data Availability

The raw data supporting the conclusions of this article will be made available by the authors on request.

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
