# Peer review of "Assessing Pragmatic Skills in People with Intellectual Disabilities"

_behavsci, 2025, doi:10.3390/bs15030281_

Round 1

Reviewer 1 Report

Comments and Suggestions for Authors

Thank you for giving me the opportunity to review this paper. Overall, I believe that the topic of this systematic review is of clinical value and the study has the potential to contribute to the existing literature. However, I think several changes are needed; I believe the focus of the review is too broad for useful conclusion to be reached. I have specific points and comments in the next sections.

L 31: “Introduction” – a) I think you should describe the ID population in more detail in the introduction because it presents with substantial diversity, b)The justification for the review should be more clear. It is not clearly explained how table 1 was compiled and what was the rationale. Is the order used significant? It includes tests targeting diverse populations, both children and adults, both with and without intellectual disabilities, and tests that do not specifically target pragmatics, PICA for instance, I think is a test for overall aphasia severity. I think you should explain the process used to develop this table more explicitly.

L. 110: I think that the specific research questions based on which you organize the presentation of the findings are not clearly stated

L. 134 “Inclusion criteria”- a)I think you should justify the decision to include both children and adult participants with ID. You do not explicitly discuss this choice and the different assement requirements for e.g. non-verbal children vs. functional adults. b) Did you take into account the language of the test and participants? How did you approach tests adapted from English? C) the use of the term pragmatics for the assessment is very broad which is evident in the heterogeneity of the papers reviewed

L. 148: “…OR ‘mental retardation’ OR ‘mental retardation’ OR ‘mental retardation’)”- Did you mean to include something different here?

L. 178; “Analysis of the studies and profile of the participants” – The information in this paragraph is not clearly related to a specific research question. The data presented seem disparate and not clearly related to one another. Variables such as the language of the participants or the type of the research are not specified. I think the research question should be framed more clearly.

L. 198: “Pragmatics analysis tools used”- In this section you describe each protocol but it is not clearly organized in terms of the information presented and the aim of the description. Again, there is no clear connection to a research question.

L. 198: “Pragmatic analysis tools used”- I think you should state which tests are stardardized and whether normative data are available.

L. 289- I think  a systematic review that covers a subset of pragmatic abilities on the population and overlaps with your study should be discussed in more detail. To clarify how it differs from your study and how your study complements the existing review.

L. 400: “Risk of bias”- What is the overall conclusion in terms of risk of bias?

L. 490-498- These are tests that target pragmatics across different populations not ID specifically, and cover many different areas of pragmatics as well.

L. 507- Pragmatics is a very broadly defined field. In this paragraph you try to categorize areas of assessment. I think this is in the right direction, but it sould be meaningful then to summarize results and assessment procedures based on these 4 pragmatic areas and possibly differentiate children from adults.

L.524- This paragraph that discusses cogntive vs. lingustic focus of the tests is not very clear. Needs to be explained in more detail.

L. 524: “…although those are behind the most frequent pragmatic errors (articles 1, 3, 4, 5, 7, 9, 12, 13, 14, 18 and 19)…” – Have you discussed/analyzed this information previously when you described the test and studies?

L. 535: What are the implications of the participant similarities discussed in this paragraph?

Minor notes:

L. 19: “pragmatic”- reword; L. 36: [6]- need to add a a period

L. 52: “unitelligibility” is not directly related to pragmatics

L. 87 – Is this premise for the CCC supported by literature? Do you have any similar data for the most frequently used pragmatics test for adults?

L. 112: …”studies where pragmatics has been specifically assessed in population with ID will be analyzed”- rephrase. I think the text should be reviewed for proofreading

L. 179: This paragraph needs proofreading

L. 211-215: “In [57] the results mostly referred that the data on socio-communicative skills ithe SW population are similar to their peers… sometimes, improved adaptive behaviours [59].”- this section should be rephrased

L. 224: “…completed their application with other tools”- rephrase

L.279: “… in children with 22q11.2DS, with ID and with ID + Autism into form, content or usage problems.” -rephrase

L. 321: …. , underlying that they discuss more than they speak.- I it is not clear wht this means

L. 336: Table 3- Is the term figure correct here?

Table 2: Information presented on the last collumn is not easy to understand

L. 482: “According to…”- rephrase

L. 509: …”appropriateness to the social situation [59628764658374];” ??

L. 561: “In conclusion, this systematic embodies a reliable source of evidence, as the entire procedure was carried out through peer review, with each researcher replicating the entire process and obtaining coinciding results” - rephrase

L. 587: “As development progresses, problems decrease or disappear [¡Error! No se encuentra el origen de la referencia.]” ??

Comments on the Quality of English Language

I think the quality of English is acceptable

Reviewer 2 Report

Comments and Suggestions for Authors

COMMENTS AND SUGGESTIONS FOR AUTHORS

I believe that the study “Assessing pragmatic skills in young people with intellectual disabilities” presents interesting findings regarding the need to design and develop assessment tools for pragmatic skills in individuals with intellectual disabilities. The review adheres to PRISMA guidelines for conducting systematic reviews. Below, I provide a series of suggestions related to both the content and format of the paper, which is generally adequate.

  1. The introduction references several tools for assessing pragmatic skills, which are listed in Table 1. However, in the text, Table 1 is only cited in relation to the Functional Communication Profile.
  2. Table 1 appears to present (if I understand correctly) the main tools for assessing pragmatic skills. It includes, for example, the Subtest of the Revised Navarra Oral Language Test (PLON-R), even though the test’s authors consider it the least reliable subscale of the test. However, more recent tools specifically focused on early detection of pragmatic difficulties, such as the EDPRA (Botana & Peralbo, 2023), are not mentioned. Moreover, if general language development tests with pragmatic subscales are included, the Clinical Evaluation of Language Fundamentals–5 (CELF–5) is missing. This section of the introduction should be reviewed.
  3. In Table 1, the format for the authors’ names in the "Author/s" column is inconsistent (sometimes using last names and initials, other times full names and last names, etc.). Additionally, “&” and “y” are used interchangeably. This should be standardized.
  4. The citations on lines 46 and 56 should follow the same format. Please review the entire text for consistency.
  5. In the search equation on line 148, OR 'mental retardation' is repeated three times.
  6. In Table 2, Study 17 does not include information on either the diagnosis or the gender of the participants.
  7. In the section on CCC-2 between lines 217 and 228, the DOI in citation 61 is not functional in the references section. A study that could complement the data presented regarding the application of this tool to children with Down syndrome is Carril & Pereira (2019): Pragmatic abilities in children with ASD, ADHD, Down syndrome, and typical development through the Galician version of the CCC-2.
  8. There is an error on line 588.
  9. In the abstract, between lines 25 and 28, the authors suggest the need to develop standardized tools for assessing pragmatic skills in individuals with intellectual disabilities. However, this aspect is not addressed in the discussion or conclusions. Do the authors still support this claim based on the results of their review?
  10. Please review the references section. Not all references follow the same format. Errors or incomplete information were identified in references 67, 78, 79, 80, 81, and 88.

Round 2

Reviewer 1 Report

Comments and Suggestions for Authors

Overall, the authors made an effort to address all queries and all changes were in the right direction. I have only a few comments: 

  1. 1st draft comment on L. 134 “Inclusion criteria”- & L. 178; “Analysis of the studies and profile of the participants” a)I think you should justify the decision to include both children and adult participants with ID. It is still not clear what was your age criteria for participant selection, you have a very broad age and this is not communicated clearly in the paper (the method or the title), also the  age range included affects and the tests used. You have 2 year old children and 40 year old adults.
  2. 1st draft comment L. 134 “inclusion criteria” & L. 178; “Analysis of the studies and profile of the participants”. Did you take into account the language of the test and participants? How did you approach tests adapted from English?: these questions were not addressed in the revised version
  3. 1st draft comment on L. 289 regarding the systematic review of Đorđević et a., should be more thoroughly addressed: although you discuss risk of bias you do not explain any further how your study differs in content and how it overlaps in the revised version

Minor comments

L 33: …develop pragmatics…: rephrase sentence

L 87:… and with Spanish language: rephrase

L 98: … to form a global coherence… rephrase

L. 181: …interactional pragmatics (60%); to assess enunciative or textual 181 pragmatics (25%): Do you explain these terms?

L.293: Finally, Đorđević et al. (2014) conducted a systematic review on the assessment of paralinguistic aspects such as emotion recognition in people with ID, revealing that they are most frequently studied in etiologies or syndromic specificities: Diagnostic Analysis of Nonverbal Accuracy Scale-2 (Nowicki & Duke, 1994), Florida Affect Battery (Bowers et al., 1991), Test of Facial Emotion Recognition (Young et al., 2002), and Self-Report Anger Inventory (Benson & Ivins, 1992). : I think you need to rephrase…

L. 402: Do you have the references for these tests?

Author Response

We attach a document with the response.
